# Developing a Non-Pharmacological Intervention Programme for Wandering in People with Dementia: Recommendations for Healthcare Providers in Nursing Homes

**DOI:** 10.3390/brainsci12101321

**Published:** 2022-09-29

**Authors:** Jing Wang, Ge Zhang, Min Min, Ying Xing, Hongli Chen, Cheng Li, Caifu Li, Hanhan Zhou, Xianwen Li

**Affiliations:** 1School of Nursing, Nanjing Medical University, Nanjing 210000, China; 2The Affiliated Brain Hospital of Nanjing Medical University, Nanjing 210000, China; 3Landsea Long-Term Care Facility, Nanjing 210000, China; 4School of Medicine, Lishui University, Lishui 323000, China; 5Clinical and Nursing Training Center, Jiangsu Health Vocational College, Nanjing 210000, China

**Keywords:** non-pharmacological interventions, programme, dementia, wandering, elderly care

## Abstract

Background: Wandering among people with dementia (PwD) is associated with a high risk of injury and death. The stigma of dementia prevents Chinese dementia families from seeking information and support earlier, which increases the demand for long-term care facilities. Despite universal recognition of the importance of care facilities, healthcare providers in care facilities still lack the relevant nursing knowledge and skills, including non-pharmacological interventions (NPIs) that have been proven to be effective in preventing wandering. Systematic and culturally appropriate NPI programmes for healthcare providers to manage wandering among PwD in long-term care facilities are still lacking. We aimed to develop an evidence-based and culturally appropriate NPI programme for wandering in PwD to guide healthcare providers in nursing homes to prevent wandering and its adverse outcomes. Methods: The NPI programme was developed according to the framework of the Belgian Centre for Evidence-Based Medicine (CEBAM). We, (1) performed a systematic literature search to summarize the available evidence, (2) developed evidence-based recommendations for the NPI programme based on the existing evidence, and (3) carried out a validation process to revise the content of the recommendations and to determine the grades of recommendations, including group meetings with experts and a survey for end-users. Results: Based on 22 publications and validation from 7 experts and 76 end users, we developed 21 recommendations covering 4 domains: (1) caregiver education, (2) preventing excessive wandering, (3) promoting safe walking, and (4) preventing people with dementia from going missing. We created almost all recommendations of the four domains with accompanying levels of evidence and grades of recommendations. Conclusions: By combining the evidence with expert and end-user opinions, a comprehensive NPI programme was developed to support institutional healthcare providers to prevent wandering and its adverse outcomes. The benefits of this programme are currently being tested.

## 1. Introduction

With the incredible speed of population ageing, someone develops dementia every three seconds, and the number of people with dementia is set to increase to 152 million by the year 2050 [1]. The overall prevalence of dementia among people aged at least 60 years in China is projected to be 6.7% in 2030, reaching 23.3 million [2]. Dementia is a syndrome characterized by the deterioration in cognitive function beyond what might be expected from the usual consequences of biological ageing [3]. In 2020, the total cost of dementia care to families and the health care system was estimated to be 248.71 billion USD in China, and this number is expected to reach 507.49 billion USD in 2030 [4]. In addition to cognitive impairment, behavioural and psychological symptoms of dementia (BPSD), also known as neuropsychiatric symptoms of dementia, are core features [5]. As the disease progresses, over 90% of people with dementia (PwD) will eventually be affected by BPSD in the form of aggression, agitation, hallucinations, delusions, wandering, and sleeping disorders [6]. BPSD increases the caregiver burden and reduces the quality of life of PwD and their caregivers, and dementia has been regarded as a public health priority [3].

Wandering, a common trait found in PwD, refers to seemingly aimless or disoriented ambulation throughout a facility, often with a wide range of behaviours such as lapping, pacing, and random ambulation [7,8]. The cause of wandering is multifaceted as it can be in response to hunger, thirst, pain, or confinement [9], or it can be associated with a specific personality trait, poorer behavioural response to stress, and greater functional and balance impairment [10]. Adekoya et al. examined the perspectives of PwD and reported that wandering was an expression of unmet needs such as a desire to be with family, relieve boredom, continue a lifelong habit, or socialise with others [11]. The exact prevalence of wandering is difficult to determine, but an estimated 25 to 63% of institutionalised PwD or those with cognitive impairment may wander at some point over the course of the disease [12]. A previous study found that 15 to 60% of all PwD and up to 25% of community-dwelling PwD exhibited wandering behaviour [9]. Although wandering within a safe environment improves appetite and provides opportunities for exercise and social contact [13], PwD who wander away from home or facilities can experience adverse outcomes such as sleep disturbance, injuries from falls and traffic accidents, getting lost, and even death [8,14]. In addition, PwD may intrude into other people’s personal space with ensuing altercations, loss of privacy, and risk of physical harm, all of which impact quality of life [11].

In order to prevent these events, effective strategies should be taken to reduce wandering and ensure the safety of PwD. Despite universal recognition of the severe impact of dementia on people’s lives, Chinese people are generally unwilling to discuss dementia and mental health issues with others, and the stigma of dementia and the current one-child family structure in China might leave PwD without adequate support at home, increasing the demand for long-term care facilities [15,16]. However, previous studies have found several major barriers in the current practice of dementia care in care homes in China, including an unfriendly environment, inappropriate care culture in care homes, and poor skills and knowledge in managing the BPSD of dementia [16,17]. An extensive literature search demonstrated that healthcare providers in the community and care homes lack the relevant nursing knowledge and capacities [18,19], and dementia-friendly environments are not yet established, resulting in the special needs of PwD not being satisfied [20]. Some care facilities refuse to admit PwD because of inadequate staff competence and service to meet the care needs [21]. Therefore, providing healthcare providers in care facilities with strategies such as the selection of electronic tracking devices, environment-based interventions, and meaningful activities is essential to prevent the adverse outcomes of wandering and to ensure the safety of PwD. 

Studies on pharmacological and non-pharmacological interventions (NPIs) for wandering have been widely carried out. As pharmacological interventions could lead to unwanted side effects and negative consequences [22], personalized NPIs play an essential role in the management of wandering. High-tech strategies, including global positioning systems (GPS), radiofrequency, and electronic tracking, can monitor a person’s exact location and alert responsible individuals [9,23,24]. Low-tech strategies, such as environment-based interventions [25,26,27,28,29] and supporting facilities [30,31], can effectively reduce wandering and night behaviours, help PwD in their wayfinding, and decrease the risk of falls. In addition, regular supervised exercise, such as gait and balance training, and strength training, can be effective in improving balance and decreasing the risk of falls [32]. Among previous studies on NPIs, some studies present syntheses of research on partial NPIs, such as environmental interventions, and several studies have explored the impact of comprehensive interventions in reducing PwD from going missing or experiencing falls. However, previous studies on NPIs for wandering in PwD still have some limitations. First of all, most studies only cover some of the NPIs for wandering and do not systematically present a synthesis of interventions to prevent wandering and adverse health outcomes. Next in importance, previous studies have failed to evaluate and improve the evidence on the management of wandering through the involvement of experts and end users. Due to extensive research in recent years, updated research on comprehensive interventions is needed. Last but not least, despite universal recognition of the importance of care facilities for PwD, culturally appropriate NPI programmes for healthcare providers in care facilities to manage wandering are still lacking in China.

Healthcare providers who care for PwD in nursing institutions require further training to increase their knowledge and capacities. Based on a systematic literature search, we aimed to evaluate and summarize the existing professional knowledge and the best evidence about NPIs for wandering in PwD to develop a systematic and culturally appropriate NPI programme. This optimized NPI programme will serve as a valuable tool to educate institutional healthcare providers on how to prevent and manage wandering and its adverse outcomes to improve the quality of life of PwD.

## 2. Materials and Methods

We used the procedure of the Belgian Centre for Evidence-Based Medicine (CEBAM) model [33] to develop the NPI programme. The CEBAM model is distinguished by government departments’ involvement in addressing socialization issues. The main procedure entails three stages: (1) a systematic review of the literature relating to NPIs for wandering in PwD, conducted to summarise the available evidence; (2) the development of recommendations for the NPI programme based on the existing evidence and opinions of the author group; and (3) a validation process to revise the content of recommendations and to determine grades of recommendations.

### 2.1. Stage 1: Literature Search

#### 2.1.1. Identification of the Research Question and Domains

We assembled a working group to develop the recommendations: a neurologist nurse, a geriatric psychiatric specialist nurse, and a geriatric specialist nurse. Collectively, they have more than ten years of experience in dementia care. Based on a previous literature review and the combined experience of the working and author groups, we identified the research question: What NPIs are currently available to help healthcare providers prevent wandering in PwD? Through discussion and consensus within the author and working groups, we derived four domains: (1) caregiver education, (2) preventing excessive wandering, (3) promoting safe walking, and (4) preventing PwD from going missing.

#### 2.1.2. Literature Search Strategy

According to the conceptual framework provided by the 6S pyramid [34], we systematically searched information resources using a four-step strategy, the details of which can be found in Appendix A, Table A1, Table A2, Table A3, Table A4 and Table A5. The search and study selection process is reported using the PRISMA flowchart [35] to improve the standardization of the writing process.

(i) To retrieve evidence-based decision information, we undertook an initial search of UpToDate and BMJ Best Practice from the inception of the project to 12 July 2021. The relevant search terms included wandering and dementia. Furthermore, we performed a systematic literature search in the Cochrane Library and the Joanna Briggs Institute (JBI) evidence-based practice to retrieve articles published between 2016 and 2021. Search terms consisted of both keywords and medical subject heading terms.

(ii) We searched existing guidelines published from 2016 to 2021 in various databases, including the National Institute for Health and Clinical Excellence (NICE), the Registered Nurses’ Association of Ontario (RNAO), and the National Guideline Clearinghouse (NGC). The relevant search terms included wandering and dementia. Further, we included appropriate guidelines when searching other databases such as PubMed. Due to the absence of pertinent guidelines for the past six years, we extended the timeframe for searches.

(iii) Since there were few intervention studies on the wandering behaviour of dementia patients, we supplemented the search of databases such as PubMed, Embase, and CINAHL. We restricted the timeframe for searches of publications from January 2016 to December 2021. We developed the search strategies according to specific database requirements and employed keywords and medical subject heading terms, which were the same throughout.

(iv) By referring to the International Alzheimer’s Association member associations [36], we searched the web for Alzheimer’s Association websites in countries where Chinese or English was the official language. Finally, we searched on the Alzheimer’s Society website, and the relevant search term was wandering. Information on the Alzheimer’s Society is shown in Table 1.

#### 2.1.3. Inclusion and Exclusion Criteria

Eligible articles had to meet the following criteria:The participants were older adults with any form of dementia.The study was about specific NPIs for wandering in PwD.Articles meeting the following criteria were excluded:The study only included pharmacological interventions.Interventions were implemented in hospitals.Conference abstracts, protocols, introductions, or reviews other than systematic ones.

#### 2.1.4. Quality Assessment of the Included Studies

Both authors, who had received a period of training beforehand, independently assessed the guidelines using the Appraisal of Guidelines and Research and Evaluation II (AGREE II) instrument [37]. The AGREE II tool comprises 23 items assessing 6 domains: (1) scope and purpose; (2) stakeholder involvement; (3) rigour of development; (4) clarity of presentation; (5) applicability; and (6) editorial independence. Each domain item is rated based on a 7-point Likert scale ranging from 1 for strongly disagree to 7 for strongly agree. Finally, there are two overall assessment items, including the overall quality of the guideline and recommendation for use. The overall assessment was based on the six domains’ scores and the authors’ judgement. To solve a difference between the two authors, a method was adopted to assess the overall quality of the guideline: If the authors assigned a score difference of 1 point, the lower score was used. If the difference was 2 points, the average value was taken. If the difference was more than 2 points, a consensus was reached after discussion [38]. Guidelines were considered high quality if 5 or more domains scored >60%, average quality if 3 or 4 domains scored >60%, and low quality if 2 or fewer domains scored >60% [39,40]. In addition, we calculated mean and standard deviation scores.

The methodological quality of the included articles was evaluated by two authors independently using the JBI Critical Appraisal Checklist for the following study types: systematic reviews [41], analytical cross-sectional studies [42], cohort studies [42], and quasi-experimental studies [43]. Each question posed in the checklist could be scored as yes, no, unclear, or not applicable. Two authors independently read and evaluated the articles, and a third author resolved any conflicts.

#### 2.1.5. Data Collection

Two authors independently used a standard data extraction form to extract the relevant data. In case of disagreements, a consensus was necessary for resolution. An overview of all included publications is provided in Table 2.

### 2.2. Stage 2: Development of the Intervention Programme

Data were extracted from each study and then used to inform the first draft of possible recommendations drawn up by the author group. The consensus method was used to establish the critical recommendations of the intervention programme. When there was disagreement regarding the evidence or rationale for/against a particular point, a majority opinion was required for resolution.

The level of evidence was evaluated using the JBI Level of Evidence for Effectiveness [58]. The JBI comprises five levels of evidence: (1) experimental designs, (2) quasi-experimental designs, (3) observational-analytic designs, (4) observational-descriptive studies, and (5) expert opinion and bench research. When levels of evidence from different sources were inconsistent, the author group followed the principle that evidence-based results came first, as well as high-quality evidence.

### 2.3. Stage 3: Validation Process

We conducted additional validation of the recommendations. The findings were discussed within the author group, and recommendations were revised if necessary and applicable.

(1) The first draft of the NPI programme was developed into a consultation questionnaire e-mailed to the experts (nursing specialists working with PwD); informed consent was also obtained. Afterwards, group meetings were organized to assess and improve each formulated recommendation. The author group modified the recommendations according to the opinions put forward by the experts and refined the NPI programme following expert validation.

(2) An online survey was set up to assess the importance and familiarity of recommendations in the first domain, and the feasibility and degree of completion of each formulated recommendation in the last three domains. The survey was then e-mailed to potential end users, including health care professionals in hospitals and institutions and family members of PwD. The respondents were asked to score recommendations on a scale of 1 to 10 for (i) importance, (ii) familiarity, (iii) feasibility, and (iv) degree of completion.

(3) The final NPI programme was identified, and the grades of recommendation were assigned by the author groups and experts using the JBI Grades of Recommendation [59]. The following key factors were considered: the balance between desirable and undesirable effects, the quality of the evidence, values and preferences, costs, and levels of evidence.

## 3. Results

### 3.1. Search Results

Based on a broad search, we identified 230 articles, 55 of which we removed because they were duplicates. Further, we included five additional papers: three by extending the timeframe, and two from the reference lists. In a restrictive analysis of titles and abstracts, we excluded 102 articles from further analysis. Next, we subjected 78 articles to full-text analysis; 56 items did not meet the inclusion criteria. Finally, 22 articles constituted the evidence and validation base upon which we developed the recommendations. The results of the search and selection process are illustrated in Figure 1.

### 3.2. Characteristics of the Included Studies

In terms of the study design, we included one guideline [28], three systematic reviews [23,27,44], two cohort studies [46,47], two quasi-experimental studies [25,26], two analytical cross-sectional studies [30,45], three clinical decision-making studies [32,50,51], one evidence summary [49], one recommended practice [48], and seven articles from the Alzheimer’s Society [31,52,53,54,55,56,57].

Some articles involved many aspects, and we broke them down to discuss them; therefore, the number of articles discussed in this section was greater than 22. Two studies [52,55] reported the antecedents of wandering, and one on the consequences of wandering [46]. Six studies focused on preventing falls and promoting safe walking, including exercise interventions, removing rugs and excessive clutter, and installing automatic nightlights [27,32,47,53,54,57]. Seven studies [25,28,31,45,48,51,55] centred on how to prevent excessive wandering, including three on managing sleep issues to prevent night-time wandering [45,51,55]. Among the 13 studies [23,26,27,28,30,31,44,48,49,50,53,55,57] on preventing patients from getting lost, low-tech strategies such as camouflaged doors and mirrors in front of exit doors, and high-tech solutions such as identification bracelets, boundary alarm systems, and GPS devices were involved, and one of them discussed ethical issues in the use of electronic tracking devices [23]. In addition, there were four studies on what to do if a person with dementia goes missing [31,50,52,56].

### 3.3. Quality Appraisal

Of the 22 eligible articles, we directly included 5; we evaluated 1 using the AGREE II instrument and appraised 9 studies using the JBI Critical Appraisal Checklist. Directly included articles encompassed three clinical decision-making studies, one evidence summary, and one recommended practice. We used the seven articles from the Alzheimer’s Society only as search clues and supplementary references. Table 2 outlines the articles. In addition, after methodological quality assessment and group discussion, we included all ten articles (evaluated using tools); the results of the methodological quality assessments are presented in Appendix B, Table A6, Table A7, Table A8, Table A9 and Table A10.

### 3.4. Expert and User Validation

There is little insight into NPIs since physicians continue to prioritize pharmacological interventions. The expert selection was mainly in the nursing field through the group discussion. In total, seven experts confirmed the recommendations and helped to improve them further. End users, including health care professionals in hospitals and institutions and family members of PwD, were asked to assess the importance, familiarity, feasibility, and degree of completion of the recommendations. The importance score of the first domain of the programme and the feasibility and degree of completion score of the last three domains were all above average, so the recommendations were not deleted. The evaluation result is shown in Table 3. The characteristics of the participants in the validation process are described in Table 4.

### 3.5. Recommendations Based on Levels of Evidence and Grades of Recommendation

We formulated 21 recommendations covering four domains: (1) caregiver education, (2) preventing excessive wandering, (3) promoting safe walking, and (4) preventing PwD from going missing. The recommendations of the four domains are presented in Table 5, with accompanying levels of evidence and grades of recommendation. If any of the included articles containing recommendations could not be assessed through the JBI Levels of Evidence, the levels of evidence could be assessed against the original literature underlying the recommendations, and recommendations that could not be assessed through the JBI Levels of Evidence were repeatedly discussed by the working and author groups to decide whether to include them in the programme. A grade of “A” implied alignment with ‘strong’ recommendations, whereas a grade of “B” suggested alignment with ‘weak’ recommendations.

### 3.6. Caregiver Education

Whereas wandering within safe limits may have some benefits, such as providing physical exercise and social contact and improving appetite [9,13], the adverse consequences of wandering may outweigh its benefits, especially when PwD wander away from home or facilities. These adverse consequences include experiencing physical injuries from falls or traffic accidents, getting lost, and death [8,9,13,46,47]. Effective interventions to keep PwD safe should be taken to prevent these adverse outcomes, but physical restraints are considered unacceptable interventions to prevent wandering due to ethical issues and negative outcomes [23,49]. As highly unpredictable events, adverse outcomes such as becoming lost can happen to all PwD, even if they have never wandered previously, and the most common situation is when PwD are unattended in their residences [26]. A study of search files found that nine PwD who went missing from home or residential facilities died, and six of them experienced unnatural deaths, including exposure and drowning [60]. Thus, efforts should be made to ensure adequate supervision to prevent risky situations [28,48]. However, it is unrealistic to maintain constant supervision. The burden on healthcare providers could be reduced by adopting appropriate strategies, such as utilising boundary alarm systems and electronic tracking devices [23,28,44,49,50,57]. Boundary alarm systems around the gate and door sensors can send an alert if someone opens it [57], and electronic tracking devices can track, record and monitor PwD, using GPS and radiofrequency [23]. However, these devices raise profound ethical questions about their use with this vulnerable population. There should be a balance between the need for protection and safety and the patient’s need for autonomy and privacy [44,57].

Getting lost is inevitable, although extensive interventions may be put in place. Healthcare providers should be educated on the response plan in advance to search for PwD who go missing in a short time period [26,28].

When PwD become lost, the following steps should be taken [31,50,52,56]:Keep calm.Contact the local police immediately and provide a recent colour photo of the missing person, a description of his or her clothes, and details about past walking experiences, favourite places, or anywhere the person may have gone.Search the house and the surrounding buildings immediately.The initial 6 to 12 h of the search should cover an eight-mile radius around the location where the lost person was last seen, concentrating on open, populated areas, including the inside of easily accessible buildings.If the missing person has not been found, intense foot searches should focus on natural and sparsely populated areas, beginning within a two-mile radius of the last known location and extending from there, and ponds, gardens, and tree lines should be carefully searched.Search strategies should not be based on personal characteristics and experiences since PwD often exhibit unpredictable behaviour when lost.If the missing person has not been found, searches should continue through the night.If PwD travelled by bus or subway, initial search efforts should focus on locating the vehicle.

### 3.7. Preventing Excessive Wandering

The cause of wandering is multifaceted as it may be an expression of unmet needs, such as a desire to relieve boredom or socialize with others [11], or it may arise due to loss of memory and confusing night with day [52,55]. Environment-based interventions may be a way to prevent excessive wandering. Previous studies found that music stimulation using preferred songs increased positive participation in PwD and was modestly effective in decreasing wandering and night behaviours if the music is not loud [27,28,48]. Keeping the environment dark during the night and bright during the day, hanging oversized clocks, and eliminating unnecessary night-time awakenings decreased the number and mean duration of wandering and excessive wandering at night [25,51]. 

Additionally, physical exercise should not be unnecessarily limited, as it may ensure PwD get enough light during the day and improve sleep quality to reduce wandering [28,45,61]. Regular supervised exercise, such as walking after meals, prevents wandering [50,62]. Whatever the reason, people are less likely to go far if they are physically tired. Identifying the time of day when PwD are most likely to wander and providing opportunities to engage in structured, meaningful activities can help reduce anxiety and agitation [31,52]. Social interactions or daily activities, such as receiving visitors, folding laundry, or preparing dinner, may decrease excessive wandering [28,48].

### 3.8. Promoting Safe Walking

As PwD get older, their vision, muscle strength, and coordination weaken, making them vulnerable to falls [54]. Providing a safe-as-possible environment for PwD could promote safe walking [48]. Environmental stressors, such as being cold at night, and changes in daily routines and furniture, should be minimised [28,30], and tripping hazards should be removed to reduce the risk of falls by PwD, such as throw rugs, extension cords, excessive clutter, and electric cords or wires [28,53]. A previous study found that home modifications and repairs were effective ways to reduce injuries and led to an estimated 31% reduction in the rate of fall-based injuries each year [63]. 

In addition to environmental interventions, regular supervised exercise can reduce the desire to wander and the risk of falls and related injuries, to promote safe walking for PwD [32,50,64,65,66,67]. Exercise as a single intervention reduced the number of falls and recurrent falls by 36% and 41%, respectively [68]. One meta-analysis suggested that programmes that involved a high challenge to maintain one’s balance and included more than 3 h per week of exercise had better fall prevention effects [69]. To ensure the safety of PwD, it is necessary to provide suitable shoes and clothes, as well as a secure place to wander and exercise, such as a lounge or garden [28,31]. Since maintaining constant supervision is unrealistic and rates of fall-related hospitalisations among PwD remain higher than among those without dementia [70], night monitoring systems, such as alarm systems, tracking devices, and automatic nightlights could be used to reduce falls [27].

### 3.9. Preventing PwD from Going Missing

For PwD with spatial disorientation, environmental cues to reduce the risk of loss and assist them in wayfinding may be effective, such as printed or graphic signage, personal items such as photographs, and posters and murals on walls [27,28]. Wayfinding cues should be salient and straightforward, and the amount of irrelevant information should be minimised [71]. Handrails in hallways following installation strategies of orientation and continuity helped PwD find their way quickly and should be mounted not much higher than the eyes due to weak necks [30].

High-tech strategies should be employed to reduce attempts at exiting. For example, monitoring systems (such as warning bells and devices above doors) tracking the patient’s position, and sending signals (such as bells and buzzers) when a door was opened [31,50,52,64,72]. The following low-tech strategies can do the same. Unobtrusive safety measures (such as camouflaged doors, horizontal grids of black tape in front of exits, safety covers, or cloth of the same colour as the door in front of exit doors) can act as visual stop barriers to reduce attempts at exiting [27,31]. Tactile boards, interactive walls, or 3D wall art with visual appeal can reduce attempts at exiting by diverting attention [28]. Moreover, PwD should not be left unsupervised or locked in a room [31]. If PwD get lost, forms of identification, including identification bracelets, clothing labels, and tracking devices, can help them get home quickly [26,73].

## 4. Discussion

Few systematic NPI programmes are available for healthcare providers in nursing homes to prevent and manage the wandering behaviour of PwD. In the present article, a comprehensive set of recommendations to provide healthcare providers with a NPI programme was developed by maintaining a systematic approach. Compared with previous studies on NPIs for wandering in PwD, the outstanding feature of the NPI programme developed in this work is that the four domains, including caregiver education, preventing excessive wandering, promoting safe walking, and preventing PwD from going missing, were sorted out by different purposes and priorities. Therefore, healthcare providers could make targeted choices. Most existing studies on NPIs for wandering are not comprehensive and systematic and only cover part of the contents of this NPI programme, such as the response plan on getting lost [31,50], appropriate environmental modifications [27,28], and supervised regular exercise [32]. In addition, very few guidelines summarise comprehensive NPIs for managing wandering in PwD and preventing adverse consequences of wandering behaviour, and no comprehensive guidelines have been published in recent years. For our programme, the first domain, including caregiver education and the development of the response plan, makes up for the lack of healthcare providers’ education from previous NPI strategies for wandering. The latter three domains combine low-tech and high-tech strategies and emphasise the role of personalised environmental modifications and different electronic devices in reducing and managing wandering and ensuring the safety of PwD. Experts point out that until there is an appropriate quantitative standard, the duration and frequency of intervention implementation should be based on the patient’s tolerance, acceptability, and financial situation. Given the low quality of evidence on NPIs for wandering, expert opinions can be one of the sources of evidence when the quality of evidence is not high, or resources are insufficient [74]. Some interventions positive for different domains were not combined into a single item in the intervention programme but instead elaborated based on different purposes, such as high-tech strategies. In addition, parts of the response plan in the first domain were revised after discussion with the experts. For example, to find PwD in a short period of time and reduce the immediate risk of death, the author group decided that police should be called as soon as PwD go missing [50] rather than after 15 to 20 minutes of searching [31,56,62]. Based on the clinical situation, the initial search area was changed from a five-mile radius to an eight-mile radius around the location where the lost person was last seen, and the intensive foot search area was changed from a one-mile radius to a two-mile radius of the last known location and extended from there [50].

Moreover, the comprehensive set of recommendations for NPIs regarding wandering in PwD in this study could be better applied in practice with the participation of experts and end users. The procedure of the CEBAM attaches great importance to the validation process with emphasis on the involvement of experts to revise recommendations of the programme if applicable [75]. For dementia care, the end users have a close relationship with the people with dementia. The clinical experience of dementia specialists and the involvement of end users make this NPI programme more relevant and effective for use in practice. Also, as the programme is based on the cultural and clinical situation in China, and cross-cultural adjustments to some recommendations were made. The NPI programme is easier for healthcare providers to implement in Chinese culture and clinical situations.

Readers must be aware of several limitations of this study. The first limitation is the relatively small number of experts (*n* = 7) who replied to the survey. Second, the primary source of recommendations is the lack of high-quality studies, such as randomised controlled trials, and there are few guidelines on wandering, with only one published in recent years. In addition, the quality of some of the articles included was not assessed formally. We recommend additional training for healthcare providers before using the programme in the future, as merely providing a programme such as this will not be enough to improve practices. The final programme requires further testing in clinical practice to evaluate its feasibility and effectiveness and should be improved based on more high-quality research.

## 5. Conclusions

As the disease progresses, the needs of PwD vary, and adults with dementia cannot receive adequate support at home, resulting in increased demand for long-term care facilities. There are few systematic and culturally appropriate NPI programmes for healthcare providers to manage the BPSD of dementia, such as wandering. This article developed a comprehensive NPI programme regarding wandering in PwD to provide culturally appropriate recommendations for healthcare providers in nursing homes. These recommendations can serve as a valuable tool to enrich their knowledge about wandering and guide them in preventing wandering and its adverse outcomes. The benefits of this programme are currently being tested.

## Figures and Tables

**Figure 1 brainsci-12-01321-f001:**
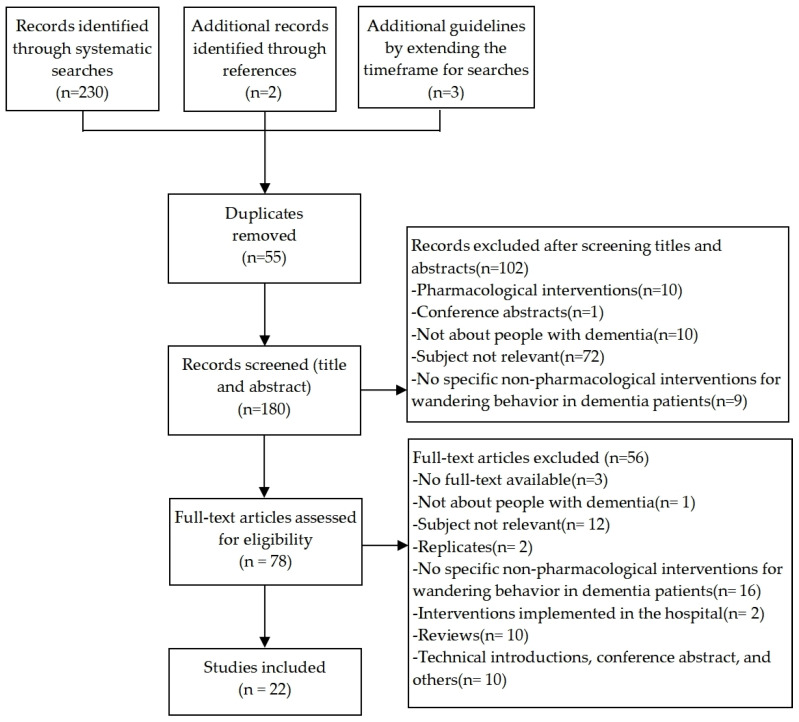
PRISMA flow diagram of the search process.

**Table 1 brainsci-12-01321-t001:** Overview of the Alzheimer’s Society (*n* = 9).

The Alzheimer’s Society	Website	Location
Alzheimer’s Disease International	https://www.alz.co.uk	The global
China Association for Alzheimer’s Disease	http://caad.org.cn	China
Dementia Australia	http://www.dementia.org.au	Australia
Alzheimer’s Association	http://alz.org	USA
Alzheimer’s Society	http://www.alheimers.org.uk	UK
Hong Kong Alzheimer’s Disease Association	http://www.hkada.org.hk	Hong Kong, China
Alzheimer’s New Zealand	http://www.alzheimers.org.nz	New Zealand
Alzheimer’s South Africa	http://www.alzheimers.org.za	South Africa
Dementia Singapore	http://alz.org.sg	Singapore

**Table 2 brainsci-12-01321-t002:** Overview and characteristics of the publications included (*n* = 22).

**Systematic Reviews (*n* = 3)**				
	**Study (Author, Year)**	**Study Type**	**Number of Publications Included (*n*)**	**Overall Appraisal ^a^**
1	Husebo et al., 2020 [44]	Systematic Review	34	Include
2	Jensen et al., 2017 [27]	Systematic Review	42	Include
3	Howes et al., 2021 [23]	Systematic Review	22	Include
**Quantitative and Experimental Research (*n* = 6)**				
	**Study (Author, Year)**	**Study Type**	**Setting (Sample, *n*)**	**Overall Appraisal ^a^**
1	Shih et al., 2017 [45]	Analytical cross-sectional study	Participants from dementia outpatient clinics of several hospitals and long-term care resource management centres in southern Taiwan (*n* = 184)	Include
2	Leung et al., 2020 [30]	Analytical cross-sectional study	Elders with dementia living in care and attention homes in Hong Kong (*n* = 65)	Include
3	Lau et al., 2019 [26]	Quasi-experimental study	Patients from a hospital-based geriatric memory clinic (*n* = 54)	Include
4	Bautrant et al., 2019 [25]	Quasi-experimental study	Patients aged 65 years or older (*n* = 19)	Include
5	Sato et al., 2018 [46]	Cohort study	People from three geriatric health service facilities (*n* = 242)	Include
6	Bowen et al., 2018 [47]	Cohort study	Older adults from a residential care facility (*n* = 26)	Include
**Guideline (*n* = 1)**				
	**Study (Author, Year)**	**Study Type**	**Overall Assessment ^b^**	
1	Futrell et al., 2014 [28]	Guideline	Recommended with modifications	
**Recommended Practice, Evidence Summary, and Clinical Decision-Making (*n* = 5)**				
	**Study (Author, Year)**	**Study Type**		
1	JBI, 2021 [48]	Recommended practice		
2	Koh, 2021 [49]	Evidence summary		
3	Daniel Press, 2021 [50]	Clinical decision-making		
4	Ariel B Neikrug et al., 2022 [51]	Clinical decision-making		
5	Douglas P Kiel, 2022 [32]	Clinical decision-making		
**Articles from the Alzheimer’s Society (*n* = 7)**				
	**Title**	**Date of Last Update**	**The Alzheimer’s Society**	
1	Wandering [52]		Dementia Australia	
2	Wandering and getting lost: Who is at risk and how to be prepared [31]	2020	Alzheimer’s Association	
3	Home Safety [53]		Alzheimer’s Association	
4	Keeping the home safe [54]	2017	Alzheimer’s South Africa	
5	Wandering [55]		Hong Kong Alzheimer’s disease association	
6	Safer Walking [56]	2019	Alzheimer’s New Zealand	
7	How technology can help [57]		Alzheimer’s Society	

**^a^** The JBI Critical Appraisal Checklist was used to evaluate the methodological quality of systematic reviews, analytical cross-sectional studies, cohort studies, and quasi-experimental studies. ^**b**^ The AGREE II instrument was used to assess the included guidelines, which comprises 23 items assessing six domains and two overall assessment items.

**Table 3 brainsci-12-01321-t003:** Descriptive analysis and comparison of feasibility and degree of completion.

	Feasibility	Degree of Completion	The Difference between Feasibility and Degree of Completion	
	M ± SD	M ± SD	t	*p*
Domain 2 Preventing excessive wandering				
6	7.47 ± 2.625	7.47 ± 2.495	0.000	1.000
7	7.18 ± 2.622	7.59 ± 2.228	–1.509	0.135
8	7.24 ± 2.993	7.41 ± 2.694	–0.568	0.572
9	8.41 ± 2.258	8.58 ± 1.982	–0.819	0.416
10	8.46 ± 2.119	8.30 ± 2.091	0.807	0.422
Domain 3 Promoting safe walking				
11	8.64 ± 2.171	8.70 ± 1.973	–0.231	0.818
12	8.63 ± 1.986	8.45 ± 2.036	0.935	0.353
13	7.80 ± 2.697	7.94 ± 2.228	–0.577	0.566
14	8.41 ± 2.264	8.50 ± 2.036	–0.357	0.722
15	7.21 ± 2.968	7.43 ± 2.754	–0.776	0.440
Domain 4 Preventing PwD from going missing				
16	7.37 ± 2.627	7.49 ± 2.615	–0.386	0.700
17	7.38 ± 2.771	7.29 ± 2.627	0.288	0.774
18	7.36 ± 2.595	7.11 ± 2.721	0.800	0.426
19	7.47 ± 2.089	7.16 ± 2.649	1.168	0.246
20	7.21 ± 2.968	7.43 ± 2.754	−0.776	0.440
21	8.36 ± 2.089	8.22 ± 2.114	0.756	0.452

**Table 4 brainsci-12-01321-t004:** Characteristics of the participants involved during the validation process.

**Experts (*n* = 7)**			
	**Professional Background**	**Working Unit**	**Seniority (Year)**
1	Clinical nurse assistant in dementia care	Institution	30
2	Social worker in geriatric and dementia care	College (Japan)	38
3	Geriatric psychiatrist	Hospital	27
4	Clinical nurse assistant in dementia care	Institution	20
5	Neurologist	Hospital	17
6	Specialist nurse in dementia care	Hospital	23
7	Specialist nurse in dementia care	Hospital	10
**End Users (*n* = 76)**			
	**Participants**	**Number of Participants (*n*)**	
1	Health care professionals in hospitals	16	
2	Health care professionals in institutions	8	
3	Family members	52	

**Table 5 brainsci-12-01321-t005:** Recommendations.

	Recommendations	Levels of Evidence	Grades of Recommendations
Domain 1	Caregiver education		
1	Wandering may provide physical exercise and social contact and improve appetite, but it can make PwD experience adverse outcomes such as physical injuries from falls and getting lost [46,47]	3c	A
2	Ensure continuous supervision to prevent risky situations, as all PwD are at risk of becoming lost, including those who have never wandered before [26,28,48,50]	2d	A
3	Physical restraint is an inappropriate intervention to prevent wandering, as it is considered to be ethically problematic [23,49]	/	B
4	Healthcare providers can choose high-tech strategies, including boundary alarm systems, monitoring systems, and electronic tracking devices for PwD, but the user’s privacy and autonomy should be respected [23,28,44,49,50,57]	1c	B
5	When PwD become lost, a response plan including the following steps should be taken [31,50,52,56]	4d	B
	- Contact local police immediately, provide information about PwD who got lost, and extend the search through social networks such as WeChat platforms and TikTok short video platforms		
	- Search the house and the surrounding buildings immediately		
	- The initial 6 to 12 hours of the search should cover an eight-mile radius around the location where he/she disappeared, concentrating on open, populated areas		
	- If initial search efforts fail, intense foot searches should focus on natural and sparsely populated areas, beginning within a two-mile radius of the location where he/she disappeared and extending from there		
	- Search strategies should not be based on personal characteristics and experiences		
	- Searches should continue throughout the night if necessary		
	- If PwD travelled by automobile or subway, initial search efforts should focus on locating his/her vehicle		
Domain 2	Preventing excessive wandering		
6	Listen to music chosen according to the patient’s preferences [27,28]	2d	B
7	Provide opportunities to engage in social interactions or meaningful activities when PwD are most likely to wander, such as folding laundry, preparing dinner, receiving visitors, or participating in live violin recitals, depending on their ability [28,31,48]	2d	B
8	Choose oversized clocks to hang in a prominent position in corridors [25]	2d	B
9	Ensuring adequate light during the day (e.g., keeping the environment bright during the day and providing regular supervised exercise, such as walking after meals) helps to reduce wandering at night [25,28,45,48,50,51,55]	1c	A
10	Keep the environment dark during the night, and eliminate unnecessary night-time awakenings (e.g., noise) [25,51]	1c	B
Domain 3	Promoting safe walking		
	Provide an environment as safe as possible		
11	- Keep the floor clean and remove tripping hazards to promote safe walking, such as excessive clutter, loose mats, and extension cords [28,48,53,54]	/	A
12	- Minimise stressors from the environment, such as changes in daily routines and furniture arrangements [28,30]	4b	B
13	Provide a secure place for PwD to exercise to reduce the risk of falls and fall-related injuries [28,31,47,52]	3c	A
	Prepare for a walk		
14	- Wear appropriate footwear and walk in the company of healthcare providers [26,50,54]	2d	A
15	- Monitoring devices should be used to prevent injuries, such as alarm systems or automatic lights [27]	1c	B
Domain 4	Preventing PwD from going missing		
	Wayfinding cues may reduce disorientation		
16	- Provide environmental cues to help PwD find their way, including photographs, posters and murals on walls, and extra-large signage, which should be salient and simple [27,28,30,48]	4b	B
17	- Handrails in hallways installed throughout the house should be oriented, continuous, and conspicuous to support dementia patients’ mobility [30]	4b	B
	Reduce attempts at exiting		
18	- Take advantage of visual stop barriers to reduce attempts at exiting, including camouflaged doors, horizontal grids of black tape in front of exits, safety covers, and cloth of the same colour as the door in front of exit doors [27,28,31]	2d	B
19	- Divert attention by using tactile boards, interactive walls, and 3D wall art [28]	2d	B
20	- Use high-tech strategies, such as warning bells above doors, monitoring systems, and tracking devices with GPS [26,44,48,50,52]	1c	B
21	Don’t leave PwD unsupervised [26]	2d	A

/Recommendations that could not be assessed through the JBI Levels of Evidence.

## Data Availability

The datasets supporting the conclusions of this article are included in the article.

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
