# Peer review of "Developing a Non-Pharmacological Intervention Programme for Wandering in People with Dementia: Recommendations for Healthcare Providers in Nursing Homes"

_brainsci, 2022, doi:10.3390/brainsci12101321_

Round 1
Reviewer 1 Report
The manuscript titled “Developing a non-pharmacological intervention programme for wandering in people with dementia: recommendations for healthcare providers in nursing homes” aimed to provide a non-pharmacological intervention (NPI) programme for wandering for People with Dementia (PwD). In order to carry out this aim, Authors a) conducted a systematic literature search on NPI programmes, b) developed evidence-based recommendations, and c) managed a validation process to consolidate them. The systematic literature search carried out 22 publications, from which Authors extracted and developed 21 recommendations covering four domains: (1) caregiver education, (2) preventing excessive wandering, (3) promoting safe walking, and (4) preventing people with dementia from going missing. Validation process was conducted by 7 experts and 76 end users. Almost all recommendations had accompanying levels of evidence. Authors discussed their results in light of previous literature and of the use to be done of the NPI programme they developed.
I carefully read the manuscript, and I think it may be of interest for the readers of Brain Sciences. I also think that some minor points need to be addressed before the manuscript could be published as a research article. Below there are my comments and suggestions.
Overall, the manuscript is very well-written and properly addresses the relevant issue of wandering in persons with dementia as well as lack of clear and established guidelines for preventing dangerous outcomes of such behavior. It is also relevant the effort Authors did in both reviewing previous literature and validating their proposal by experts as well as end users. I found that the introduction section, the aims of the study and the methodology employed are clear and detailed, as well as the explanations provided in the discussion section. I only have few remarks:
Materials and Methods section
Lines 118-123, page 3: The procedure developed by the Belgian Centre for Evidence-Based Medicine (CEBAM) was used to develop the programme. Please, provide a motivation for this choice and indicate a reference about it.
Lines 173-175, page 4: The Appraisal of Guidelines and Research and Evaluation II (AGREE II) was used to assess the guidelines. Please, provide a reference about it.
Results section
Line 238, page 6: Authors used the PRISMA flow diagram for reporting the search process. What about the PRISMA guidelines for the systematic literature search? The Authors should consider and mention them also in the Materials and Methods section.
Discussion section
A recap of the results of the study was presented in the discussion section. It would be useful to add a paragraph which offers a comparison of Authors findings with the most recent and available evidence. Moreover, I would also answer to the question “What is the added value of this work with respect to previous literature and guidelines for wandering behaviour in PwD?”
Reviewer 2 Report
This paper contains a literature search in addition to new recommendations for how healthcare providers can prevent wandering among dementia patients in nursing homes. The research question is novel, and the paper should be of interest to a large number of readers. Here are some things that would improve the paper:
1) Abstract (Background): I would clarify the first sentence by saying, "Wandering among dementia patients..." (line 18). In lines 24-25, I would specify, "...to manage wandering among PwD in LTC facilities."
2) Abstract (Conclusion): I would clarify at the end of the Conclusion that, "The benefits of this program are currently being tested." (lines 38-39)
3) Introduction (line 47): Dementia is not one illness, but a variety of diseases, including Alzheimer's disease, vascular dementia, dementia with Lewy bodies, frontotemporal dementia, etc. Therefore, the sentence that ends in "...core features of the illness [3]" does not make too much sense.
4) Introduction (lines 47-49): "As the disease progresses..." should be changed to, "As these diseases progress, over 90% of PwD will eventually be affected by BPSD.....[4]."
5) Introduction (lines 103-106): This is a run-on sentence. Consider ending the first sentence with, "...the involvement of experts and end-users," and beginning the next sentence with, "Due to extensive research...".
6) Introduction (line 114): There is no need for 2 adjectives. I would eliminate the word, "multi-perspective", and simply say, "This optimized NPI program...."
7) Methods (Table 4): I would be more specific about the characteristics of the participants involved in the validation process. For example, persons 1, 4, 6 were listed as "Dementia care". Were these clinical nurse assistants (CNAs)? Person 2 was listed as "Geriatrics". Was this a geriatric physician? geriatric nurse? geriatric social worker? Person 7 was listed as "Cognitive impairment". Was this a neuropsychologist? a patient?
Reviewer 3 Report
Dear Authors,
I read your work entitled "Developing a non-pharmacological intervention programme for wandering in people with dementia recommendations for healthcare providers in nursing homes" and my suggestion to you is to have a more thorough and a in depth edit in English and syntax errors because in some cases a reader can understand what you want to state but it does expressed in clear and sound way.
Your work is in full details and has a great merit.
Thank you.
Reviewer 4 Report
The aim of the current study was to develop an evidence-based and culturally appropriate preventive programme for wandering in people with dementia, which can guide healthcare providers in nursing homes to prevent wandering and its adverse outcomes. The programme was developed in three steps. First, a systematic literature review on non-pharmachological interventions for wandering in PwD was conducted, then specific recommendations based on the existing evidence and expert opinions were developed and, finally, the recommendations were validated. The overall process is appropriate and easy to replicate. I do not have any concerns about the methodology of the paper.
